# A different perspective for nonphotochemical quenching in plant antenna complexes

Edoardo Cignoni [1], Margherita Lapillo[1], Lorenzo Cupellini [1✉], Silvia Acosta-Gutiérrez [2], Francesco Luigi Gervasio [2,3✉] & Benedetta Mennucci [1✉]

Light-harvesting complexes of plants exert a dual function of light-harvesting (LH) and photoprotection through processes collectively called nonphotochemical quenching (NPQ). While LH processes are relatively well characterized, those involved in NPQ are less understood. Here, we characterize the quenching mechanisms of CP29, a minor LHC of plants, through the integration of two complementary enhanced-sampling techniques, dimensionality reduction schemes, electronic calculations and the analysis of cryo-EM data in the light of the predicted conformational ensemble. Our study reveals that the switch between LH and quenching state is more complex than previously thought. Several conformations of the lumenal side of the protein occur and differently affect the pigments' relative geometries and interactions. Moreover, we show that a quenching mechanism localized on a single chlorophyll-carotenoid pair is not sufficient but many chlorophylls are simultaneously involved. In such a diffuse mechanism, short-range interactions between each carotenoid and different chlorophylls combined with a protein-mediated tuning of the carotenoid excitation energies have to be considered in addition to the commonly suggested Coulomb interactions.

[1] Dipartimento di Chimica e Chimica Industriale, University of Pisa, via G. Moruzzi 13, 56124 Pisa, Italy. [2] Department of Chemistry, University College London, WC1E 6BT London, UK. [3] School of Pharmaceutical Sciences and ISPSO, University of Geneva, CH-1211 Geneva, Switzerland. ✉email: lorenzo.cupellini@unipi.it; francesco.gervasio@unige.ch; benedetta.mennucci@unipi.it

Light-harvesting complexes (LHCs) in the photosystems of plants and green algae use aggregates of chlorophylls (Chls) to collect sunlight and transfer excitation energy to the photosystem core. However, an excess of light can be detrimental to such a delicate photosynthetic apparatus, and photoprotective mechanisms are needed to avoid photodamage[1,2]. The main strategy is to dissipate the energy absorbed by chlorophylls into harmless heat; this is made possible through processes collectively called nonphotochemical quenching (NPQ)[3–8].

The ways through which such dissipation proceeds are still debated. Most authors agree that the carotenoid molecules (Cars) embedded in LHCs actively participate in the quenching[9–11]. Within this hypothesis, the most widely accepted mechanism for the quenching involves excitation energy transfer (EET) from the excited Chls to the dark ($S_1$) state of Cars[9,12]. A second mechanism has also been proposed which entails a charge transfer (CT), whereby the excited Chls accept an electron from the Cars[10,11,13,14].

Recent experimental studies also suggest that these quenching processes are mediated by conformational changes of the LHCs[15–19]. As a matter of fact, LHCs can access various conformational states that present specific spectral and kinetic properties, but their connection to quenched and unquenched states is still to be revealed[16,20–22]. Moreover, while it is clear that NPQ is triggered by a pH decrease in the lumenal side of the membrane and the following activation of an additional protein, the PsbS protein[2,22], the mechanisms through which PsbS interact with the LHC and affects its state are not known.

In this work, we focus on a minor LHC of higher plants and green algae, CP29 (Fig. 1). At variance with the main antenna complex, LHCII, CP29 is a monomeric complex, found at the interface between the peripheral antenna complexes and the core complexes of Photosystem II. Its central position makes CP29 an ideal site for photoprotection[23]. CP29 binds 13–14 chlorophylls and three carotenoids (Fig. 1b), namely lutein (Lut), violaxanthin (Vio), and Neoxanthin (Neo)[24,25]. The binding sites of Lut (L1) and Vio (L2) have both been related to the photoprotective function[19,26,27]. In particular, it is thought that Lut and Vio can quench the excitation of the tightly associated Chls $a$612 and $a$603[28], respectively. Both of them have been related to an excitation-energy sink, the "terminal emitter" domain[9,19,29]. In both cases the quenching is believed to go through an EET which is determined, according to the Förster theory, by the Coulomb interactions between excitations localized on the Chl and the Car, respectively. Within this model, in order to shift from the active to the quenched state, a significant change in Chl–Car interactions (e.g., in the value of their Coulomb coupling) is necessary. This, in turn, implies a large modification of the relative distance and/or orientation of the two interacting pigments which should follow the conformational change of the complex.

Here, a different perspective is suggested for the regulation of NPQ activity in CP29. By combining two complementary enhanced sampling techniques[30,31] with unbiased molecular dynamics simulations[32,33] and dimensionality reduction methods, we thoroughly explore the conformational landscape of the complex and establish how the apoprotein conformation impacts the L1 and L2 sites and the interactions among the embedded pigments. These effects are quantified in terms of both long-range Coulomb couplings and short-range effects. Our analysis reveals that the hypothesis of a switch between an active and a quenched state is too simplistic, and instead, several conformations of the lumenal side of the protein occur and differently affect the pigments relative geometries and interactions. Moreover, we show that a description based solely on the Coulomb interactions within the most coupled Car–Chl pairs cannot explain the photoregulation. Instead, a more diffuse mechanism involving all the pigments within both L1 and L2 sites is possible. In such a mechanism, short-range interactions between each carotenoid and different Chls, combined with a protein-mediated tuning of the carotenoid excitation energies, have to be considered.

## Results

**Exploration of protein conformations.** The conformational sampling of CP29 is here enhanced by employing parallel tempering in the well-tempered ensemble (PT-WTE)[34,35]. This approach, which was successfully used in a number of systems[36,37], leads to a faster exploration of the conformational landscape of the CP29 complex. In this way, free energy barriers of moderate size can be overcome without explicitly introducing a collective coordinate to bias the protein structure or the pigments towards specific

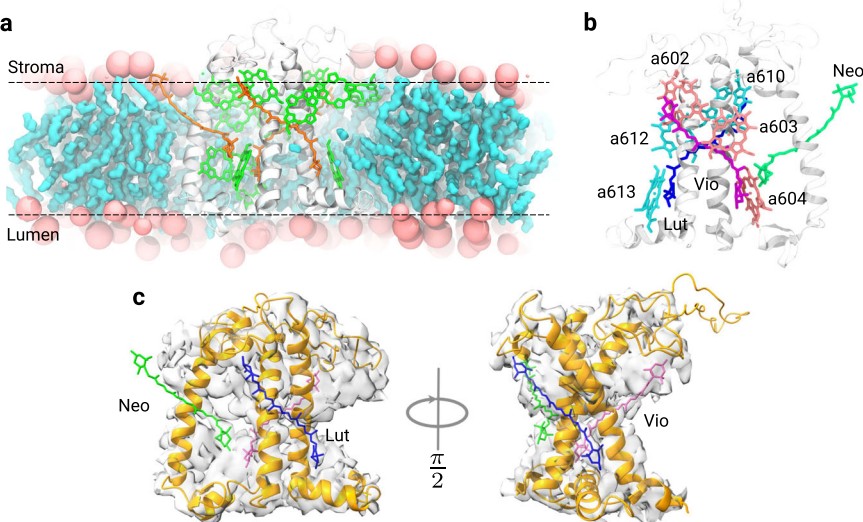

**Fig. 1 The CP29 antenna complex. a** CP29 embedded in the phospholipid membrane employed in our simulations. The lipid membrane is cut for better visualization of the complex. **b** The carotenoids and some of the chlorophylls embedded in CP29: Lut (blue) in L1, Vio (purple) in L2, and Neo (green) in N1. Only the Chls in sites L1 and L2 are shown and highlighted in cyan and pink, respectively. **c** CP29 cryo-EM structure (PDB: 3JCU) embedded in the cryo-EM electron density.

conformations. Details on the simulation protocol are provided in the Supplementary Methods. Increasing evidence[4,17,21] indicates that LHCs possess a highly heterogeneous conformational ensemble, whose equilibrium can be tuned by PsbS, the establishment of a pH gradient, and aggregation[20,22]. To capture and disentangle the conformational heterogeneity of CP29 we apply a principal component analysis (PCA) on the backbone and sidechain dihedrals (dPCA[38]) of the protein residues. The low-dimensional projection of the CP29 dynamics onto the first two dPCA principal components is shown in Supplementary Fig. 2 and compared with an unbiased simulation of CP29 (cMD$_{CryoEM}$) employed in previous work by some of us[39]. From this comparison, it is clear that the PT-WTE enhanced sampling approach explores a larger portion of conformational space, in both the stromal and lumenal sides of the complex.

The enhanced conformational freedom of the stromal part of the protein (Supplementary Fig. 2) arises from the high degree of flexibility of the N-terminal domain of the protein, which protrudes outside the membrane in our simulations (Fig. 1a). This observation is in line with previous reports.[24,40] Indeed, the first 87 residues in the N-terminal domain outside the membrane are so flexible in the isolated CP29 that their crystal structure could not be determined.[24] The situation is different when the CP29 is embedded in the much larger biological PSII

supercomplex as the N-terminal domain anchors CP29–CP47. In this case, the flexibility of the region is significantly less pronounced, as shown by the cryo-EM electron density (see also below).[41,42]

Conversely, the observed enhanced plasticity of the lumenal side of the protein should not be affected by the contacts of CP29 in the PSII and it is in line with previous suggestions that the conformational changes triggering the onset of quenched conformations take place on this side of the LHCs[43–45]. As discussed below, marked flexibility of this side of the protein is also observed in the electron density of the whole PSII complex obtained by cryo-EM.[41] Thus, we first focus our analysis on the lumenal side of the complex, and in particular on the region surrounding helix D.

The PT-WTE lumenal free energy landscape is characterized by different minima exhibiting similar stability (Fig. 2a) on the projection variables. The broadest one contains the structure reconstructed from cryo-EM data (PDB:3JCU[41]) and crystallography[24]. In order to better characterize protein conformations, we have grouped similar structures together with a hierarchical clustering algorithm. We have identified six clusters (Fig. 2b, d), corresponding to six free-energy minima in the dPCA space. Details on the clustering procedure and on the selection of the clusters are provided in the Supplementary Methods.

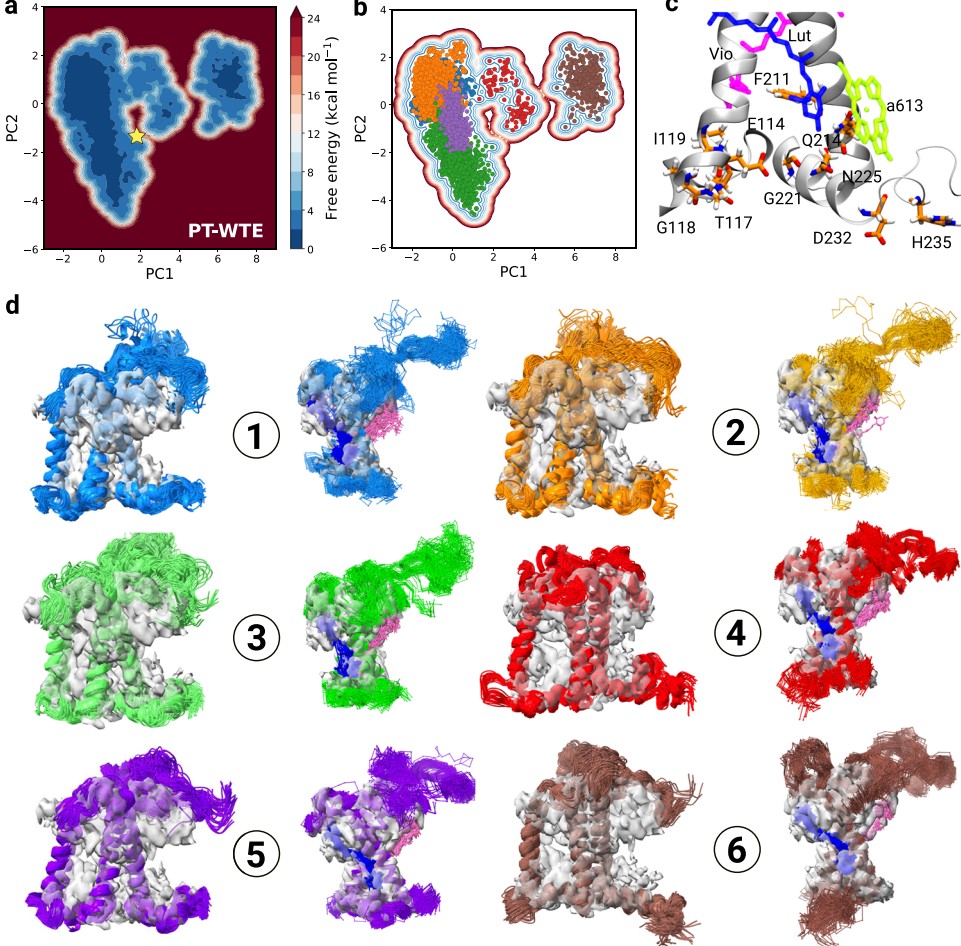

**Fig. 2 Conformational landscape of CP29 explored in the PT-WTE simulation. a** Exploration of the PT-WTE simulation of CP29, shown in the space of the first two dPCA principal components. The position of the cryo-EM starting structure (PDB: 3JCU) is shown as a yellow star. **b** Clustering of the PT-WTE simulation, visualized in dPCA space. Points colored differently correspond to different clusters. **c** Most relevant residues in the dynamics of CP29 on its lumenal side (see text). The carotenoids are shown for reference, as well as Chl *a*613 whose axial ligand is Q214. **d** Visualization of the structures sampled in the PT-WTE simulation. The structures are fitted in the cryo-EM density so as to better visualize their difference with the starting structure. The ID of the cluster is indicated in each circle.

Figure 2b shows the distribution of the clusters in the first two dimensions of the dPCA space, and in Fig. 2d random samples of a few structures per cluster are shown as cartoons with the cryo-EM[41] electron density overlaid for comparison. Clusters 4 and 6 correspond to the two well-separated minima in the free energy surface (Fig. 2a) and show significant differences in the protein backbone at the lumenal side. The remaining clusters are closer together, showing a similar backbone structure (Fig. 2d) and resembling the conformations explored by an unbiased simulation started from the cryo-EM CP29 structure (Supplementary Fig. 2). Nonetheless, these clusters can be distinguished upon examination of the third PC (Supplementary Fig. 3) and by looking at the distributions of their sidechain conformations (Supplementary Fig. 7a). In order to simplify the following discussion, the analyses are reported for the most extreme clusters (4 and 6) and for cluster 5, which well represents the free-energy basin associated with the starting cryo-EM protein conformation.

The most striking difference observed in clusters 4 and 6 is represented by an alternative position and orientation of helix D. Indeed, compared to the other helices, helix D presents the highest B-factor in the cryo-EM data[41] (Supplementary Fig. 9). Furthermore, the main helices are very well defined in the electron density, including the N-terminal of the complex, but the C-terminal is less well resolved (Fig. 1c), suggesting that this helix was present in multiple orientations in the cryo-EM dataset used for the structure reconstruction. In agreement with the cryo-EM data, we observe high mobility of helix D, which sticks out from the electron density in all the clusters. In addition, in clusters 4 and 6 helix D is in a markedly different position, farther from helix E and slightly shifted toward the lumen (Supplementary Fig. 8). In agreement with Ioannidis et al.[44], we find that the displacement of helix D observed in clusters 4 and 6 is associated with a different backbone geometry of G221. This enhanced flexibility is also in line with the findings of ref. [46], where an expansion of the lumenal side of the complex was observed for LHCII monomers, as opposed to the trimeric LHCII.

In addition to G221, several other amino acids are found to be important in determining the identity of the different protein conformations explored by our simulation (Supplementary Fig. 6), and some of them are shown in Fig. 2c. They are all localized on the lumenal side of the protein, with the exception of F211 and Q214. Interestingly, Q214 is the axial ligand of Chl $a$613, while F211 is localized near the lumenal ring of Lut, suggesting that conformational changes at the level of the protein residues do occur at the level of the L1 pocket, as recently suggested[47]. In addition, several residues (G221, N225, D232, and H235) appear

to play a role in determining the orientation and the geometry of helix D and the C-terminus. Finally, some residues (E114, T117, G118, and I119) are localized on the lumenal side of helix B and on the loop that links helix B to helix E, possibly affecting the geometry of helix E and the shape of the L2 pocket.

**Protein conformation affects pigment geometries and interactions.** Given the conformational changes observed in the PT-WTE simulation, it is of interest to see if they do impact the L1 and L2 sites and the embedded pigments. It has been reported that the pigments' interactions are extremely sensible to an alteration of their environment[48,49], both at the level of the relative intermolecular arrangement and at the level of the pigments' internal geometries. The latter point is especially subtle for carotenoids[45], for which minor changes in the geometry can significantly influence their electronic structure and, indirectly, the quenching processes[50,51]. Indeed, our simulations show that carotenoid conformations differing in their conjugated chain geometry are present in CP29 (Supplementary Fig. 11), in agreement with the results of Liguori et al. on LHCII[45]. The most pronounced changes are observed for the first conjugated chain dihedrals on the lumenal side of Lut and the stromal side of Vio, so that the carotenoids can shift between s-cis and s-trans conformations via a pedaling mechanism (Supplementary Fig. 11a, c). The conformational freedom of the two carotenoids found in the classical MD simulations was confirmed by QM/MM geometry optimizations which retained the same s-cis and s-trans distributions (Supplementary Fig. 11e). Furthermore, we observe that different dihedral distributions are associated with different clusters, supporting the view that the protein scaffold directly contributes to the establishment of different carotenoid geometries in its L1 and L2 pockets (Fig. 3a and Supplementary Fig. 11d). As variations in the carotenoid conjugated chain are expected to have an effect on its excited state properties, this suggests that small conformational changes at the level of the L1 and L2 pockets can tune the $S_1$ energies of the embedded carotenoids, as previously proposed[39,45], resulting in a modulation of the energy transfer processes mediated by the protein scaffold. We have further investigated the dependence on the geometry of the Car $S_1$ state by means of semiempirical configuration interaction (SECI) calculations, which have been recently shown to reasonably describe the electronic structure of keto-carotenoids[52]. Indeed, SECI calculations confirm the tunability of the Car $S_1$ energy, which is different in different clusters (Fig. 3c and Supplementary Fig. 5a, b) and further depends on the s-cis/s-trans conformation (Fig. 3b and Supplementary Fig. 5c, d), thereby

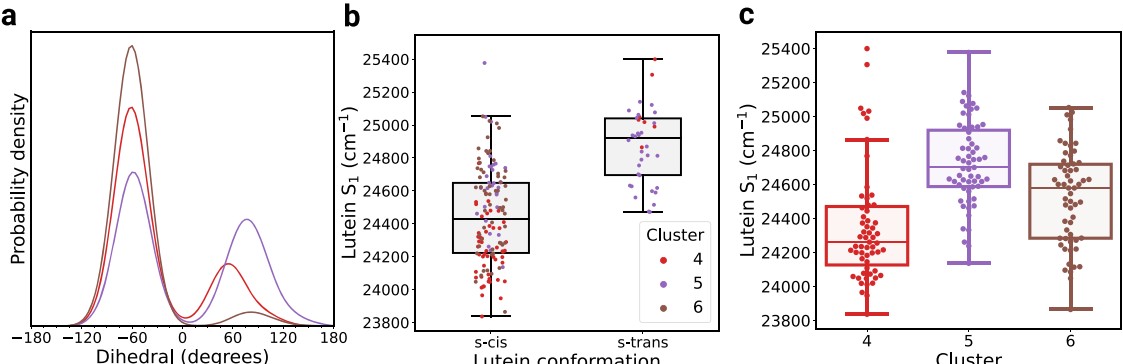

**Fig. 3 Impact of the Lutein geometry on its electronic structure. a** Distribution of the first lumenal dihedral d1$_l$ of Lutein in clusters 4–6. A precise definition of this dihedral is presented in Supplementary Fig. 11. **b** Box plot of the Lutein S1 state according to the s-cis and s-trans conformation of Lut. **c** Box plot of the Lutein S1 state in clusters 4–6. Each box plot shows the median (horizontal line), the first and third quartile (extension of the box), and the whiskers extend up to 1.5 IQR below and above the first and third quartile. The points from which each box plot is made are shown for clarity. For both **b** and **c**, $n$ = 59, 59, and 58 points from independent quantum chemical calculations are reported for clusters 4–6, respectively.

confirming the impact of the protein dynamics on the electronic structure of the embedded carotenoids.

In addition to internal conformational changes of the carotenoids, the quenching pathways strongly depend on Car–Chl interactions. The EET quenching by Car $S_1$ states has been understood in the framework of Förster's theory, where the energy transfer rate depends on the squared electronic coupling and the spectral overlap between Chl emission and Car absorption. Moreover, the electronic coupling is commonly approximated as the Coulomb interaction among transition densities of the pigments[28,53], which can be projected onto atomic charges[54].

First, we focus the analysis on the chlorophylls with the strongest coupling to the carotenoids in L1 and L2, namely Chl $a612$ with Lut and Chl $a603$ with Vio in L2. These chlorophylls lie respectively at the interface to the external (with LHCII) and the internal (with CP47) parts of PSII[55], and belong to two groups of strongly coupled chlorophylls ($a610$–$a611$–$a612$ and $a603$–$a609$) that have been proposed to form the lowest energy excitons in CP29[29], and thus have been related to quenching. As shown in Fig. 4a, our simulations indicate a remarkable insensitivity of the Coulomb coupling of Lut–$a612$ to alterations of the L1 binding pocket, with the corresponding distributions all very similar and comparable with the cMD$_{CryoEM}$ (see also Supplementary Fig. 10), in agreement with recent work[39,56]. On the other hand, the Coulomb coupling of Vio–$a603$ (Fig. 4a) appears to be quite more variable, with the distributions in each cluster covering a wide energy range. Variations of this coupling are due to the high conformational freedom of the N-terminal domain, which interacts with the stromal ring of Vio, inducing large amplitude motions of the stromal side of this carotenoid within L2 (Fig. 4c). We note that, according to a previous study by some of us[39] where a kinetic model of EET was applied to CP29, these coupling values are compatible with a quenched conformation of the antenna. We can therefore conclude that, within this Coulomb approximation of the coupling, the two most coupled Car–Chl pairs in CP29 establish an efficient quenching channel that cannot be modulated by the protein environment.

The Coulomb approximation to the EET coupling is certainly valid if the energy transfer is between bright states; however, this is not the case here where the dark $S_1$ state of Cars is involved. In these cases, and even more in triplet energy transfer (TET) where the Coulomb coupling is zero, short-range terms play a role.[57–59] Unfortunately, short-range terms are difficult to compute, as they involve charge-transfer configurations[60]. Here the difficulty of the calculations is further increased due to the lack of an established QM method for accurately describing these interactions in the case of the dark $S_1$ state.[61]

However, if we recognize that the short-range terms are strongly dependent on the overlap of electron densities of the pigments, we can obtain a rough estimate by computing the short-range contribution for the coupling between Chl $Q_y$ and the bright state of Cars ($S_2$) which is instead accurately described by time-dependent DFT approaches (see Supplementary Methods). Moreover, in a previous study of some of us,[62] we showed that the Car–Chl TET couplings in CP29 are strongly sensitive to the overlap parameter, a geometrical approximation of the electronic overlap in terms of rigid spheres centered on the atoms of the Car–Chl pair (see Supplementary Methods). In fact, the short-range contribution to the coupling behaves similarly for different singlet states[59]. Indeed, the short-range couplings calculated for the $S_2/Q_y$ energy transfer are similar in magnitude to the ones previously calculated for TET (and to the Coulomb $S_1/Q_y$ couplings reported in Fig. 4)), and they show a similar dependence on the overlap parameter as shown in Supplementary Fig. 4.

Following all these findings, we are confident that the same overlap parameter can be used here to capture the short-range character of the $S_1/Q_y$ coupling. The calculated overlap distributions do differ among the clusters (Fig. 4d). This result, when compared with the insensitivity of the Coulomb interaction

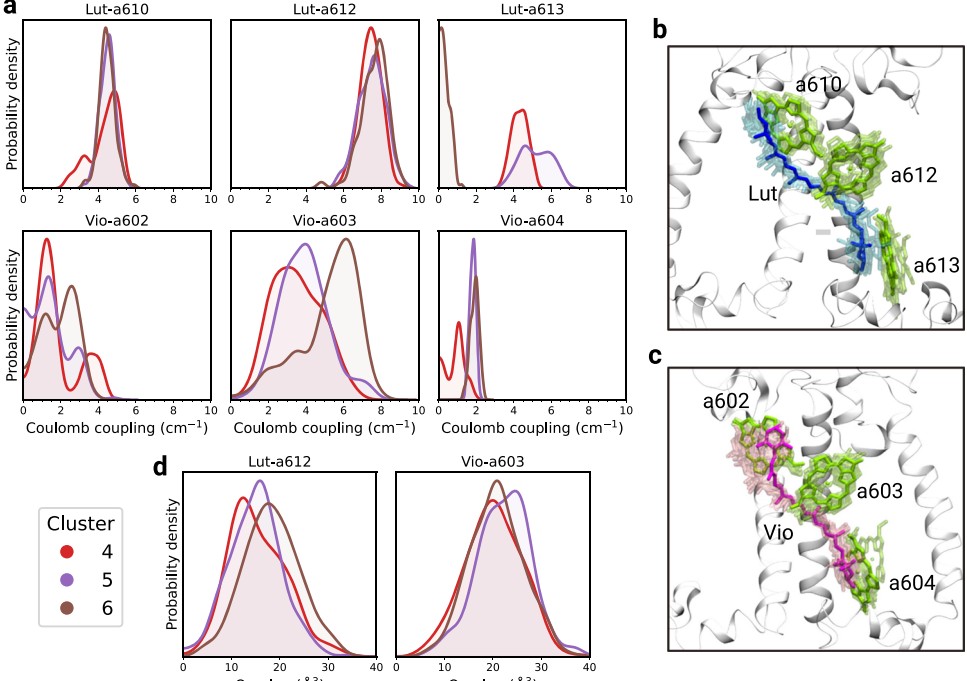

**Fig. 4 Coulomb coupling and overlap within CP29.** The distributions are shown for clusters 4–6 of the PT-WTE simulation. The density domain is bound to positive values only. **a** Distributions of the absolute value of the Coulomb coupling between Lut and Chl $a610$, $a612$, $a613$ (L1) and between Vio and Chl $a603$, $a603$, $a604$ (L2). **b** Fluctuations of Lut–Chl pairs within the L1 site. **c** Fluctuations of Vio-Chl pairs within the L2 site. **d** Distributions of the overlaps for the pair Lut–$a612$ (L1) and Vio–$a603$ (L2). The distributions for all clusters and for the cMD$_{CryoEM}$ simulation are shown in Supplementary Fig. 10.

observed in L1 (Fig. 4a), suggests that a putative Lut–a612 quenching channel can only be modulated by controlling the short-range coupling, rather than by altering the Coulomb coupling. Variations in the overlap are also found for Vio–a603 (Fig. 4d), indicating that short-range interactions are likely to play a role also for this Car–Chl pair. Still, the short-range contributions for Vio–a603 are likely less impactful than for Lut–a612, as the increased conformational freedom of Vio in L2 (Fig. 2d, Fig. 4c) allows for sensible variations of its Coulomb coupling, as opposed to Lut (Fig. 4a).

Having found such an impact of short-range contributions to the Car–Chl interactions, it is interesting to investigate whether the alternative quenching channel, namely the one involving a charge transfer (CT) state, can be active within CP29. Indeed, CT quenching has a short-range character that has been linked to the overlap parameter in previous work by some of us[14], where it was shown that quenching in LHCII can proceed via CT state in the Lut–Chla612 pair in site L1. Analogously to what has been done for LHCII, we apply Marcus' theory to CP29 in order to estimate the CT rates (details on the analysis protocol can be found in the Supplementary Methods and in ref. [14]). Contrary to what is found for LHCII, no CT quenching channel appears to be active in CP29 (Supplementary Table 1). This agrees with what is found experimentally, as CT quenching in CP29, if present, has been reported only for Zeaxanthin-binding CP29[10,63]. The inactive CT quenching channel within CP29 is the result of the higher energy of the CT state found in both the L1 and L2 sites (Supplementary Table 1). On the basis of the results of ref. [14], the lower energy of the CT state in LHCII can be rationalized by considering the additional stabilization of this state by means of a positively charged Lysine residue (Lys179 in LHCII), which is substituted by an Alanine in CP29.

For both sites, we observe larger differences in the coupling distributions when considering the chlorophylls at the two extrema of Lut (Chl a610 and a613) and Vio (Chl a602 and a604) (Fig. 4a), indicating that Car–Chl interactions on the two sides of the carotenoids are easier to tune than for the central Chl, thereby suggesting the presence of multiple Car–Chl quenching channels. In particular, Chl a613, which is anchored to helix A via Q214, shows the largest coupling displacement especially in clusters 4 and 6 (Fig. 2d), indicating that the mobility on the lumenal side of the protein particularly affects this Chl. On the other hand, the Vio–Chl a602 interaction is affected by the conformational freedom of Vio in the stromal side (Fig. 4c).

**A metastable "Open" conformation is observed in the lumenal side of CP29.** The above analysis indicates that conformational changes in CP29 can affect the geometry of the L1 and L2 sites. Still, it seems that altering the L1 site has almost no effect on the Lut–a612 Coulomb coupling. This implies that, without taking the short-range contributions of the coupling and the modulation of the site energies into account, it is not possible to tune the interactions of the Lut–a612 pair; however, extensive work suggests that an EET quenching channel at the level of this pair is indeed present[19,26]. It is then interesting to investigate if, by enhancing the sampling of more distorted structures of CP29, it is possible to observe a variation in the Lut–a612 coupling, thereby indicating that major distortions of the protein can effectively alter the Car–Chl interactions at the level of their Coulomb coupling. To this end, we follow refs. [46,64,65] where it was proposed that an alteration of the inter-helical crossing angle between helices A and B (the torsion defined by the nodes A2–A1–B1–B2 in Fig. 5a) could account for NPQ in LHCII monomers. In addition, given the enhanced plasticity observed on the lumenal side of CP29, we have split the P1 angle of ref. [64] in a

lumenal P1 (P1$_l$) and a stromal P1 (P1$_s$), so as to better describe the two sides of the complex (Fig. 5a). The sampling over P1$_l$ and P1$_s$ is enhanced with well-tempered metadynamics[66] in its multiple walkers[67] variant and successively refined with well-tempered metadynamics on path CVs[68]. Details on the simulation protocols are provided in the Supplementary Methods.

By enhancing the sampling over P1$_l$ and P1$_s$ an additional basin appears as a metastable state in the free energy surface (Fig. 5b). This CP29 conformation differs significantly from the cryo-EM[41] structure and from the conformations sampled in the cMD$_{CryoEM}$ and PT-WTE simulations, in particular for a wider inter-helical angle on the lumenal side (Fig. 5c) of the protein. For this reason, we refer to this conformation as Open. The metastability of this conformation has been further confirmed with two 2 $\mu$s long unbiased simulations, to which we refer collectively as cMD$_{Open}$. We estimate that this conformation is separated by several kcal mol$^{-1}$ from the cryo-EM one (Fig. 5b, Supplementary Fig. 12), and additional stabilizing factors would then be required to make it accessible at room temperature.

The wider P1$_l$ angle observed in Open is obtained by a large amplitude motion of the lumenal side of helix A, which additionally drags helix D away from helix B (Fig. 5c). This conformational change has a direct influence over the L1 site, which appears to be significantly altered (Fig. 5d). In particular, we find both a different position for the chlorophylls in the L1 site and a distorted geometry for Lutein, which keeps its interactions with helix D despite the departure of the latter. Finally, a softer effect is found for the L2 site (Fig. 5g), due to the greater stiffness of helix B compared to helix A.

Given the pronounced alterations of the L1 site, it is natural to expect a variation in the Lut–a612 interactions in the Open conformation. However, we again observe a remarkable insensitivity of the Coulomb coupling, which is lowered by a very small (~7%) extent in the Open conformation (Fig. 5e). It appears that even large rearrangements of the protein scaffold are not sufficient to modulate the Lut–a612 Coulomb interactions, suggesting that this Car–Chl interaction has little weight on the modulation of NPQ.

Instead, in agreement with the PT-WTE results (Fig. 4a), the Vio–a603 coupling is quite variable (Fig. 5h) due to the high amplitude motions of the N-terminal domain of CP29. Variations in the Coulomb coupling are also found for the chlorophylls on the two extrema of the carotenoids (Chls a610, a613 for Lut and Chl a602 for Vio, Supplementary Fig. 13), with the coupling of Lut–a613 being the most affected, due to the remarkable plasticity of helix A (Fig. 5a). As noted before for the PT-WTE simulation, it appears that the main role of the protein scaffold dynamics is to alter the carotenoid geometry (Fig. 5d, g), with direct consequences on its electronic structure and excited-state properties. Finally, we note that the Open conformation has an effect also on the overlap distribution of Lut–a612 (Fig. 5f), which decreases with respect to the PT-WTE results. Also, in this case, short-range interactions, very sensitive to the overlap between Car and Chl, can indeed be strongly tuned by the conformational change.

## Discussion

It is generally accepted that conformational changes of LHCs are responsible for their switching between a light-harvesting and a quenched state. Furthermore, there have been several studies suggesting that Chl–Car EET based on a Coulomb approximation of the coupling can be used to explain the quenching[28,53,64,69,70]. However, there seems to be a difficulty in understanding how the quenching efficiency can be regulated within the dynamic environment of these antennas.[39] Indeed, albeit an efficient quenching channel is mandatory for NPQ to work, this same

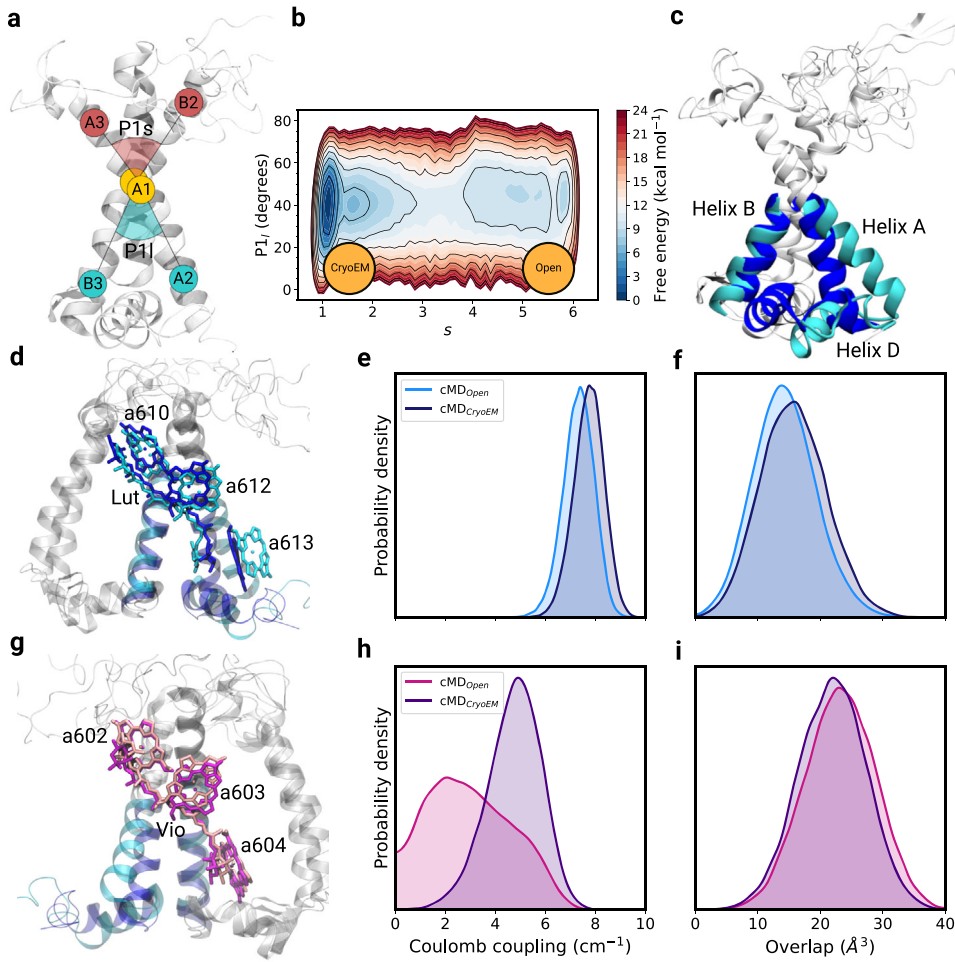

**Fig. 5 Metadynamics simulation and Open structure of CP29. a** Definition of $P1_l$ and $P1_s$. **b** Free energy surface in the space of the path CV $s$ and $P1_l$. The cryo-EM basin corresponds to small values of $s$, while the Open structure corresponds to values of $s$ comprised between 5 and 6. **c** Comparison of the cryo-EM structure (PDB: 3JCU, dark blue) and the Open structure (cyan). **d** Alteration of the L1 pocket when moving from the cryo-EM structure (dark blue) to the Open structure (cyan). **e** Coulomb coupling and **f** overlap for the Lut–$a$612 pair. **g** Alteration of the L2 pocket when moving from the cryo-EM structure (magenta) to the Open structure (pink). **h** Coulomb coupling and **i** overlap for the Vio–$a$603 pair.

channel should be switched on and off by different protein conformations, allowing plants to dissipate or collect light in response to external stimuli. What is then the connection between protein conformational changes and the regulation of NPQ?

Here, by using enhanced sampling techniques, we have been able to explore the conformational space of CP29, identifying various conformers differing in their lumenal arrangement. We have shown that these conformers correspond to different arrangements of the L1 and L2 sites both in terms of the internal geometry of the Cars and the Car–Chl relative disposition. Notwithstanding this structural flexibility of sites L1 and L2, the Coulomb interactions of the Cars with the central chlorophylls ($a$612 and $a$603) do not show very large variations. In particular, the Coulomb coupling of the most coupled pair, Lut–$a$612, is surprisingly insensitive to protein conformational changes even when largely altered structures of CP29 are considered.

This demonstrates that the Coulomb contribution to the EET coupling results in a quenching channel that cannot be tuned by the LHC conformation, indicating that this popular model is not sufficient to explain the NPQ regulation, but additional aspects must be added. The first is the inclusion of short-range effects, which become important for the closely associated Car–Chl pairs such as Lut–$a$612 in L1 and Vio–$a$603 in L2. Indeed, our estimates of the short-range effects indicate that only by adding this

contribution to the coupling can the Car–Chl interaction becomes more sensitive to the protein conformation, thus linking the modulation of the quenching efficiency to the dynamic nature of LHCs. The second aspect to consider is the effect of the protein conformation on the Car internal geometry. In fact, our results indicate that both Lut and Vio undergo geometrical distortions in the respective sites, which are reflected in a significant tuning of the $S_1$ energy and, consequently, of the overall dynamics of the antenna complex. This indicates that the control of the Car geometry is one way for the protein to tune the quenching efficiency[45], without significantly altering the Chl network. Finally, our results indicate that the couplings between the Cars and the external chlorophylls of sites L1 and L2 cannot be neglected and they are sensitive to protein conformational changes. This suggests that multiple Car–Chl EET quenching channels are active, and can be regulated by different protein motions. This underlines the importance of considering all the Chl–Car interactions in L1 and L2 sites to understand the excitation quenching in antenna complexes.

All the findings of our study refer to a CP29 isolated from the rest of PSII. However, the electron density of the complex reconstructed by cryo-EM strongly indicates that the mobility observed in the isolated complex is conserved in the photosystem. Thus, the connection here revealed between the protein

conformation, the structure of the pigment aggregate, and the quenching mechanisms specific to the complex per se are expected to be valid in the context of the PSII supercomplex. However, the conditions used for the present simulation and for most in vitro studies are different from those existing in vivo. In particular, the external triggers for NPQ activation, such as the proton gradient across the thylakoid membrane and the interaction with PsbS, are not present. It is possible that those triggers can change the relative stability of the different conformers (including the metastable Open one), for instance by promoting protein-protein interactions.

Finally, it is important to recall that due to structural similarity and sequence homology, the mechanisms proposed here are expected to be relevant for other light-harvesting complexes. The atomistic details here revealed can represent a starting point for the design of complexes with optimized photoprotective functionalities.[71–73]

## Methods

We optimized the carotenoid geometry by first selecting a subset of frames from clusters 4–6, by applying farthest point sampling to obtain a representative sample of each cluster. The carotenoid geometry was optimized at the B3LYP/6-31G(d) level of theory within a QM/MM ONIOM scheme. The QM part contained only the carotenoid, while the MM part is described with the same parameters employed in our MD simulations. MM residues within 6 Å; from the carotenoid were allowed to move during the geometry optimization. The $S_1$ energy of Lut and Vio was obtained with SECI calculations. The orbitals employed in the SECI calculations were determined through an SCF calculation on an open-shell singlet state with two singly occupied orbitals. SECI calculations are based on the OM2 Hamiltonian. The CT energy, the coupling with the CT state, and the full coupling between the Car $S_2$ and Chl $Q_y$ were obtained with a TDA/$\omega$B97X-D/6-31+G(d) level of theory, in which the effect of the environment is included at MMPol level, followed by a diabatization procedure through our multi-FED–FCD scheme, analogously to what is done in ref.[14]. Additional details are provided in the Supplementary Methods. Details on the unbiased trajectory for CP29 started from the cryo-EM (cMD$_{CryoEM}$) and from the Open (cMD$_{Open}$), the setup of the PT-WTE simulation, and the well-tempered metadynamics protocols are also provided in the Supplementary Methods. Further information on the dimensionality reduction, the clustering of the conformations in the PT-WTE simulation, the Coulomb coupling calculation, and the role of the overlap in describing the short-range quenching are also given.

**Reporting summary**. Further information on research design is available in the Nature Research Reporting Summary linked to this article.

## Data availability

The datasets generated during and/or analyzed during the current study are available in the Zenodo repository, https://zenodo.org/record/5703113.

## Code availability

The custom code used for this study is available from the corresponding authors upon request.

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

## Acknowledgements

E.C., L.C., and B.M. acknowledge funding by the European Research Council, under the grant ERC-AdG-786714 (LIFETimeS). M.L. acknowledges funding by the University of Pisa under the grant BIHO. B.M. acknowledges financial support from MIUR through the PRIN 2017 (grant 201795SBA3_002). F.L.G. and S.A.G. acknowledge PRACE and the CSCS (project s1046) for supercomputing time.

## Author contributions

B.M. and F.L.G. acquired funding; E.C., L.C., and S.A.G performed the PT-WTE simulations; E.C., M.L., and L.C. performed the unbiased simulations, the metadynamics simulations, and data analysis; S.A.G. performed the alignment and comparisons to the existing cryo-EM structure; E.C. and L.C. performed the QM/MM calculations; E.C., L.C., F.L.G., and B.M designed the research; all authors wrote and edited the paper. All authors approved the final version of the paper.

## Competing interests

The authors declare no competing interests.
