## [Peer Review File · Nature Communications]

A different perspective for nonphotochemical quenching in plant antenna complexesREVIEWER COMMENTS

Reviewer #1 (Remarks to the Author):

The manuscript by Cignoni et al. reports on the non-photochemical quenching mechanism of higher plants and it gives insight into the dynamics of CP29 - the minor antenna of Photosystem II - by employing elaborate computational enhanced sampling approaches. It builds on previous studies of LHCII, CP29 to do so. In line with ref Chem. Commun., 2020,56, 11215-11218 it supports the "shift of the focus from the extensive conformational sampling to the question of accurately describing inter-pigment interactions" for Light Harvesting Complexes (LHCs). However, it provides further evidence that to properly characterize the Chlorophyll-Carotenoid (Chl-Car) Excitation Energy Transfer (EET) processes within LHCs, it is necessary to include both the short-range effects and the Car's internal structure changes due to protein conformational transitions. Therefore, a noteworthy result of the study is the explicit track down of the flaws of previous approaches in terms of the inter-pigment description at the computational level. The work will be of significance to the field, and it will aid in the development of related computational approaches. The work supports the conclusions and enough details are given to reproduce the results for CP29, or possibly for other LHCs as well.

In my opinion, some issues must be addressed prior to the consideration for publication:

(1) The authors state that "The low-dimensional projection of the CP29 dynamics onto the first two dPCA principal components is shown in Figure 2, and compared with a previous unbiased simulation of CP29 (cMDCryoEM)". I am not sure I can see the actual comparison. What does a 'previous unbiased simulation' mean? 'previous' to this study? Moreover the authors in several parts of the main manuscript refer to Figure 2 in general, but instead it would be more informative to refer to specific panels of Figure 2 (e.g. Figure 2a, 2b etc).

(2) The authors refer to Figures 5, 6, 7 or 8 that are only shown in the Supplementary Information (SI). In page 5 (at the end of the first paragraph) Figure 4 might also refer to a Figure 4 in the SI and not Figure 4 in the main manuscript. This is somewhat confusing.

(3) The group has published another study on the major LHCII from spinach (Nature Communications volume 11, Article number: 662 (2020)), reporting on the possible Charge-Transfer (CT) character of the quenching mechanism that involves Chl-a 612/ L1 within LHCs. An efficient CT mechanism for quenching is reported therein. Given the access of the group to the related software, scripts and methodologies employed therein, have the authors tried to study/ probe such a CT character for the Chl-a 612/ L1 pigment pair in CP29, related to quenching, for the structures reported in the current study?

(4) Given the helix-D flexibility in CP29, do the authors observe any particular changes in Chl 614 which is anchored to helix-D? In my opinion, not only the Car's internal structure, but also changes in the Chl macrocycle structure might affect quenching. Could also Chl 613/ 614 orientational changes affect the quenching mechanism? The authors might want to comment on that.

Overall this can be a very interesting contribution.

Reviewer #2 (Remarks to the Author):

The manuscript by Cignoni et al. presents computational multiscale simulations of an important light harvesting complex (LHC), CP29. In particular, the manuscript presents the non-photochemical quenching (NPQ) mechanism in a "new-light". An impressive array of state-of-the-art enhanced sampling computational techniques were used to understand the conformational dynamics of the protein scaffold, and the results were compared with the cryo-EM results. Atomistic insights are presented in the context of the critical role of the protein scaffold in NPQ. The study offers a significant contribution to the field and has the potential to serve as guide in methodological aspects in future work on related systems. Some issues/comments to be considered are listed below:

1. It is shown that the Car's internal structure is heavily perturbed by the protein, however this also depends quite heavily on the classical parametrisation of the Car. Can the authors verify the hypothesis regarding the torsional behaviour, maybe by comparing the results with QM/MM?
2. How were the pH differences in the stroma and lumen incorporated in the model in terms of the protonation state of the titratable residues? What kind of effect does this have on the dynamics and subsequently on the NPQ?
3. What contribution does the flexible N-terminal chain (stromal part, first 87 residues) have on the current results? In the "real" intact complex, this part of the protein makes stable contact with the CP47 antenna. Any significant contribution towards the NPQ would be an undesired consequence of extracting the samples out of the native biological organisation and any physical insights from the isolated systems probably are not so useful. It would be great if the authors can comment on this.
4. I do not find the electronic analysis of the coupling fully convincing. There seems to be a mixture of QM methods involved (TD-DFT-B3LYP for Chls, DFT-MRCI for Cars) plus an empirical rescaling. This appears too empirical, with possible errors (or error cancellation), that is hard to evaluate. In addition, the transferability of the empirical method to estimate short-range effects from the previous study to the present context is questionable. It would greatly add to the credibility of the study if these aspects are considered more carefully.

Reviewer #3 (Remarks to the Author):

Review of "A new perspective for nonphotochemical quenching in plant antenna complexes" by Cignoni et al.

In this work, authors explore how protein motions affect coulomb couplings and short-range coulomb interactions, which have consequences in the determination of energy transfer quenching channels.

The use of enhanced sampling techniques is very robust and apparently necessary to make a correlation between realistic finite temperature motions and the properties of excitation energy transfer.

This is an interesting work that sets the standard on how to think about NPQ in a dynamical context. That is, one needs to explore how all pigments contribute to NPQ due to collective protein motions, which are often ignored even in standard equilibrium MD simulations.

Before considering the work further, authors need to establish the connection between what makes NPQ channels efficient and the strength and the variability of the properties they set out to compute. In other words, given a car-chl pair, what makes that an efficient channel for NPQ? Is sensitivity of Coulombic interactions as a function of collective protein motions good or bad to establish and NPQ channel? What about absolute strength of the couplings? I elaborate on these general questions in the following bullet points:

- 1) In Figure 3a they show the distribution of coulomb couplings from the enhanced MD simulations. Authors draw conclusions based on the degree of variability. First, by looking at these plots, it is not obvious to me the degree of variability and sensitivity of the coulomb coupling, between a carotenoid and the chls and between cars. These figures should be redone using the same range for all cases. I would use a range between -2.5 to 10 cm⁻¹ for all cases. This will clearly show the relative strengths of the couplings and the spread of the distribution among the clusters.
- 2) There is no interpretation about the strength of the couplings. Is it not relevant to compare their averages in order to determine optimal NPQ channels? I am sure it is.
- 3) Authors must clearly explain their hypothesis as to what makes a more efficient EET. Is it a more variable (larger distribution of couplings, coulomb or short range) or a less sensitive distribution? What about the strength of the average couplings? I am failing to make the connection between sensitivity and variability and the efficiency in quenching. This is crucial, authors must lay out their hypothesis from

the start. Besides, none of the analysis considers the resonances needed between the excited states of Chls and the S1 state of Car.

4) When I look at the couplings for Lut-a612 I do see that couplings are insensitive with respect to all possible cluster motions. I find this quite interesting. It says that no matter how the protein moves at room temperature, the Lut-a612 always delivers the same coupling. But from what I read, authors seem to imply that this is in detriment of assigning that pair as a quenching channel because it cannot be modulated: "Thus, the differences observed in Figure 3d could give rise to significant variations in the total coupling. This observation, combined with the insensitivity of the Coulomb interaction observed in L1 (Figure 3a), suggests that a putative Lut-a612 quenching channel can only be modulated by controlling the short-range coupling, rather than by altering the Coulomb coupling" I do not see the logic here.

5) Authors use the overlap as a correlated proxy to the short-range coupling, and mention that a 10 Å reduction is related to an order of magnitude drop in the coupling. This comes from a previous work by the authors where they analyzed triplet state couplings. It is not clear that this same correlation applies to singlet state couplings. Please elaborate on possible similarities/differences. Let's say that a similar correlation exists for the singlets. Are authors claiming that protein motions can access short-range couplings that are much larger than the coulomb couplings?

6) At the top of page 7, do authors mean the Coulomb coupling of Vio-a602 instead of Vio-a603? Right after, authors talk about the stromal ring of Vio, but the stromal ring of Vio appears to interact mainly with a602.

7) Speaking about variability of couplings, would it not have been equally important to analyze the variability of their DFT/MRCI S1 states? After all, if there is no resonance between the S1 state of cars and the Qy of Chl's there is no EET.

8) Figure 7b shows the free energy map as function of collective variables that produce a local minimum at an open conformation. However, it looks that the open conformation is 10 kcal/mol higher than the CryoEM, how could this conformation be present at equilibrium?

José A. Gascón

Reviewer #1:

The manuscript by Cignoni et al. reports on the non-photochemical quenching mechanism of higher plants and it gives insight into the dynamics of CP29 - the minor antenna of Photosystem II - by employing elaborate computational enhanced sampling approaches. It builds on previous studies of LHClI, CP29 to do so. In line with ref Chem. Commun., 2020,56, 11215-11218 it supports the "shift of the focus from the extensive conformational sampling to the question of accurately describing inter-pigment interactions" for Light Harvesting Complexes (LHCs). However, it provides further evidence that to properly characterize the Chlorophyll-Carotenoid (Chl-Car) Excitation Energy Transfer (EET) processes within LHCs, it is necessary to include both the short-range effects and the Car's internal structure changes due to protein conformational transitions. Therefore, a noteworthy result of the study is the explicit track down of the flaws of previous approaches in terms of the inter-pigment description at the computational level. The work will be of significance to the field, and it will aid in the development of related computational approaches. The work supports the conclusions, and enough details are given to reproduce the results for CP29, or possibly for other LHCs as well.

Authors' Reply: We thank the Reviewer for the positive comments.

In my opinion, some issues must be addressed prior to the consideration for publication:

(1) The authors state that "The low-dimensional projection of the CP29 dynamics onto the first two dPCA principal components is shown in Figure 2, and compared with a previous unbiased simulation of CP29 (cMDCryoEM)". I am not sure I can see the actual comparison. What does a 'previous unbiased simulation' mean? 'previous' to this study? Moreover, the authors in several parts of the main manuscript refer to Figure 2 in general, but instead it would be more informative to refer to specific panels of Figure 2 (e.g. Figure 2a, 2b etc).

Authors' Reply: Unfortunately, the submitted manuscript presented a formatting problem: the "Supplementary" prefix to all our references referring to the Supplementary Figures was missing. We apologize for this issue and the misunderstandings it caused. We have now added the "Supplementary" prefix to all our references to the Supplementary Figures.

Indeed, the comparison is shown in the Supplementary Information file, more specifically in Supplementary Figure 2. The cMD_{CryoEM} simulation is the same employed in a previous work by some of us (Lapillo et al., *Biochimica et Biophysica Acta (BBA) - Bioenergetics* (2020), **11**, 1861). In order to clarify this point, we have modified the main text as follows:

*"The low-dimensional projection of the CP29 dynamics onto the first two dPCA principal components is shown in Supplementary Figure 2 and compared with an unbiased simulation of CP29 (cMD_{CryoEM}) employed in a previous work by some of us (Lapillo et al., *Biochimica et Biophysica Acta (BBA) - Bioenergetics* (2020), **11**, 1861)."*

(2) The authors refer to Figures 5, 6, 7 or 8 that are only shown in the Supplementary Information (SI). In page 5 (at the end of the first paragraph) Figure 4 might also refer to a Figure 4 in the SI and not Figure 4 in the main manuscript. This is somewhat confusing.

Authors' Reply: As explained in the previous point, this issue was caused by the "Supplementary" prefix missing from all our references to the Supplementary Figures in the main text. As the Reviewer correctly suggests, our references to Figures 5, 6, 7 and 8, as well as the reference to Figure 4 in page 5, are

references to the Supplementary Figures, and not to the Figures in the main text. We have now added the “Supplementary” prefix to all our references to the Supplementary Figures.

(3) The group has published another study on the major LHCII from spinach (Nature Communications volume 11, Article number: 662 (2020)), reporting on the possible Charge-Transfer (CT) character of the quenching mechanism that involves Chl-a 612/ L1 within LHCs. An efficient CT mechanism for quenching is reported therein. Given the access of the group to the related software, scripts and methodologies employed therein, have the authors tried to study/ probe such a CT character for the Chl-a 612/ L1 pigment pair in CP29, related to quenching, for the structures reported in the current study?

Authors’ Reply: Following the suggestion of the reviewer, we ran CT calculations in CP29, and we found that the energy of the CT state for Lut-a612 in the L1 site is higher than in LHCII. This is due to the absence of a Lys residue (Lys179 in LHCII) which is substituted by an alanine in CP29. We have added a full new section in the Supplementary Information describing how we have conducted the CT analysis, and reporting a table summarizing the results. We have added a section to the main text commenting on these results:

“Having found such an impact of short-range contributions to the Car-Chl interactions, it is interesting to investigate whether the alternative quenching channel, e.g. that involving a charge-transfer (CT), can be active within CP29. Indeed, CT quenching has a short-range character that has been linked to the overlap parameter in a previous work by some of us [14], where it was shown that quenching in LHCII can proceed via CT state in the Lut-Chla612 pair in site L1. Analogously to what has been done for LHCII, we apply Marcus theory to CP29 in order to estimate the CT rates (details on the analysis protocol can be found in the Supplementary Methods and in Ref.14). Contrary to what is found for LHCII, no CT quenching channel appears to be active in CP29 (Supplementary Table 1). This agrees with what is found experimentally, as CT quenching in CP29, if present, has been reported only for Zeaxanthin-binding CP29[10,63](CP29-Zea). The inactive CT quenching channel within CP29 is the result of the higher energy of the CT state found in both the L1 and L2 sites (Supplementary Table 1). On the basis of the results of Ref.14, the lower energy of the CT state in LHCII can be rationalized by considering the additional stabilization of this state by means of a positively charged Lysine residue (Lys179 in LHCII), which is in CP29 is substituted by an Alanine.”

(4) Given the helix-D flexibility in CP29, do the authors observe any particular changes in Chl 614 which is anchored to helix-D? In my opinion, not only the Car's internal structure, but also changes in the Chl macrocycle structure might affect quenching. Could also Chl 613/ 614 orientational changes affect the quenching mechanism? The authors might want to comment on that.

Authors’ Reply: Chl b614 is only resolved in the X-ray structure (PDB: 3PL9, lacking the N-terminus), while the Cryo-EM structure (PDB: 3JCU, with the N-terminus) has the additional Chl a616. As here we start from the Cryo-EM structure, our model does not contain Chl b614. We have added a sentence in the supporting information that underlines this difference:

“We note that this structure lacks the Chl b614, which is instead resolved in the X-ray structure [3] (PDB code: 3PL9).”

We note that, being a Chl b, the absence of this chlorophyll from our model should not give different results from what would be obtained in a model including it. In fact, we concentrate on describing the

NPQ, where the excitation energy has already been transferred to the low-lying excited states of chlorophylls a.

We have already commented in the text about the importance of considering the “lateral” chlorophylls of sites L1 and L2, of which Chl a613 is the most prominent example due to its connection to helix D motions.

Reviewer #2 (Remarks to the Author):

The manuscript by Cignoni et al. presents computational multiscale simulations of an important light harvesting complex (LHC), CP29. In particular, the manuscript presents the non-photochemical quenching (NPQ) mechanism in a "new-light". An impressive array of state-of-the-art enhanced sampling computational techniques were used to understand the conformational dynamics of the protein scaffold, and the results were compared with the cryo-EM results. Atomistic insights are presented in the context of the critical role of the protein scaffold in NPQ. The study offers a significant contribution to the field and has the potential to serve as guide in methodological aspects in future work on related systems.

Authors' Reply: We thank the Reviewer for the positive comments.

Some issues/comments to be considered are listed below:

1. It is shown that the Car's internal structure is heavily perturbed by the protein, however this also depends quite heavily on the classical parametrisation of the Car. Can the authors verify the hypothesis regarding the torsional behaviour, maybe by comparing the results with QM/MM?

Authors' Reply: Following the suggestion of the reviewer, we performed geometry optimizations of the carotenoids within the protein using a QM/MM approach and we found that the dihedrals after optimization remain close to the pre-optimization values, thus confirming the validity of the conformational space described by the MM force field.

In the revised manuscript, we have added a section in the Supplementary Information explaining how the optimizations were conducted. We have added a section in the main article commenting on these results:

"The conformational freedom of the two carotenoids found in the classical MD simulations was confirmed by QM/MM geometry optimizations which retained the same s-cis and s-trans distributions (Supplementary Figure 11e)."

2. How were the pH differences in the stroma and lumen incorporated in the model in terms of the protonation state of the titratable residues? What kind of effect does this have on the dynamics and subsequently on the NPQ?

Authors' Reply:

In our model we have not included pH differences (as noted in the Conclusions). The protonation state of titratable residues was determined by an electrostatic model in our previous work (Lapillo et al., *Biochimica et Biophysica Acta (BBA) - Bioenergetics* (2020), 11, 1861) and found identical to previous estimations (Muh et al., *Phys. Chem. Chem. Phys.* 16 (2014) 11848). We note, however, that the different conformations of CP29 which are expected to characterize LH and quenched states are not generated by the pH gradient but they are specific of the complex. This is also clear by the fact that experiments on CP29 alone, where a pH gradient is not present, still reveal conformational switches. Once said that, however, it is certainly possible that differences in pH could affect the relative energy of the different conformational states. This effect has not been considered in our simulations.

3. What contribution does the flexible N-terminal chain (stromal part, first 87 residues) have on the current results? In the "real" intact complex, this part of the protein makes stable contact with the CP47 antenna. Any significant contribution towards the NPQ would be an undesired consequence of extracting the samples out of the native biological organisation and any physical insights from the isolated systems probably are not so useful. It would be great if the authors can comment on this.

Authors' Reply: We agree with the Reviewer that isolated CP29 might have a different behavior from CP29 embedded in PSII. However, we point out that the experiments are conducted on isolated CP29, where the N-terminus is free to move. In our cluster analysis we explicitly considered only the luminal part to separate different conformations of the complexes. Therefore, the distinct conformations that we analyze present differences in the luminal part. The N-terminus does have an influence on part of the L2 site, mainly on the Vio-a602 pair. However, it does not affect the L1 site.

We note that our conclusion "the Coulomb interactions of the Cars with the central chlorophylls [...] do not show very large variations" is not affected by this point, since the N-terminus flexibility is expected to enhance the fluctuations rather than reduce them.

4. I do not find the electronic analysis of the coupling fully convincing. There seems to be a mixture of QM methods involved (TD-DFT-B3LYP for Chls, DFT-MRCI for Cars) plus an empirical rescaling. This appears too empirical, with possible errors (or error cancellation), that is hard to evaluate. In addition, the transferability of the empirical method to estimate short-range effects from the previous study to the present context is questionable. It would greatly add to the credibility of the study if these aspects are considered more carefully.

Author's Reply: Our choice of methods was dictated by the difficulty of describing the S_1 state of Cars, which requires a multireference description (in this case, the transition charges were estimated at the DFT-MRCI level). In our previous work (Lapillo et al., *Biochimica et Biophysica Acta (BBA) - Bioenergetics* (2020), 11, 1861) we have compared the DFT-MRCI and RASSCF transition charges for the Lutein, which give very similar couplings. Other works use multireference semiempirical methods to describe the Car S_1 state. We apply a scaling factor, as defined in our previous work, to be consistent with the semiempirical AM1-MRCI method of ref. [69]. However, our conclusions regarding the lack of variability in couplings are not affected by the scaling factor used.

Regarding the short-range effects, we have improved our methods by computing the short-range contribution of the Qy/S2 couplings. We took advantage of the fact that the distance dependence of the short-range coupling is independent of the singlet state considered, even when the Coulomb couplings behave very differently (Hsu, *Acc. Chem. Res.* (2009), 42, 509). Our short-range estimates based on singlets and on triplets share a similar dependence on the geometrical overlap, and also have similar magnitude. We added the following paragraphs to the Results section in the light of these new results:

"The Coulomb approximation to the EET coupling is certainly valid if the energy transfer is between bright states; however, this is not the case here where the dark S_1 state of Cars is involved. In these cases, and even more in triplet energy transfer (TET) where the Coulomb coupling is zero, short-range terms play a role.⁵⁷⁻⁵⁹ Unfortunately, short-range terms are difficult to compute, as they involve charge-transfer configurations⁶⁰. Moreover, here the difficulty of the calculations is further increased due to the lack of an established QM method for accurately describing these interactions in the case of the dark S_1 state.⁶¹

However, if we recognize that the short-range terms are strongly dependent on the overlap of electron densities of the pigments, we can try to get a rough estimate by computing the short-range contribution for the coupling between Chl Q_y and the bright state of Cars (S₂) which is instead accurately described by Time-Dependent DFT approaches (see Supplementary Methods). Moreover, in a previous study of some of us,⁶² we showed that the Car-Chl TET couplings in CP29 are strongly sensitive to the overlap parameter, a geometrical approximation of the electronic overlap in terms of rigid spheres centered on the atoms of the Car-Chl pair (see Supplementary Methods). Indeed, the short-range couplings calculated for the S₂/Q_y energy transfer are similar in magnitude to the ones previously calculated for TET (and to the Coulomb S₁/Q_y couplings reported in Figure 4)), and they show a similar dependence on the overlap parameter as shown in Supplementary Figure 4.

Following all these findings, we are confident that the same overlap parameter can be used here to capture the short-range character of the S₁/Q_y coupling finding that the calculated overlap distributions do differ among the clusters (Figure 4d)."

Reviewer #3 (Remarks to the Author):

Review of “A new perspective for nonphotochemical quenching in plant antenna complexes” by Cignoni et al. In this work, authors explore how protein motions affect coulomb couplings and short-range coulomb interactions, which have consequences in the determination of energy transfer quenching channels. The use of enhanced sampling techniques is very robust and apparently necessary to make a correlation between realistic finite temperature motions and the properties of excitation energy transfer.

This is an interesting work that sets the standard on how to think about NPQ in a dynamical context. That is, one needs to explore how all pigments contribute to NPQ due to collective protein motions, which are often ignored even in standard equilibrium MD simulations.

Authors' Reply: We thank the Reviewer for the positive comments.

Before considering the work further, authors need to establish the connection between what makes NPQ channels efficient and the strength and the variability of the properties they set out to compute. In other words, given a car-chl pair, what makes that an efficient channel for NPQ? Is sensitivity of Coulombic interactions as a function of collective protein motions good or bad to establish and NPQ channel? What about absolute strength of the couplings? I elaborate on these general questions in the following bullet points:

1) In Figure 3a they show the distribution of coulomb couplings from the enhanced MD simulations. Authors draw conclusions based on the degree of variability. First, by looking at these plots, it is not obvious to me the degree of variability and sensitivity of the coulomb coupling, between a carotenoid and the chls and between cars. These figures should be redone using the same range for all cases. I would use a range between -2.5 to 10 cm^{-1} for all cases. This will clearly show the relative strengths of the couplings and the spread of the distribution among the clusters.

Author's Reply: We thank the Reviewer for this suggestion. We changed the Figure and employed for all couplings a range spanning from 0 to 10 cm^{-1} . We have also mirrored the negative values of the coupling and used a kernel density estimation with a lower bound set to 0 cm^{-1} , in order to represent the distribution of absolute Coulomb couplings. We are allowed to do this because the EET rate does not depend on the sign of the coupling. We have further decided to simplify the comparison within Figure 4 of the revised manuscript (Figure 3 of the original manuscript) by considering only the most diverse clusters (clusters 4, 5, and 6). The complete plots are provided in the Supplementary Information (Supplementary Figure 10). We have added the following sentence to the revised manuscript:

“In order to simplify the following discussion, the analyses are reported for the most extreme clusters (4 and 6) and for cluster 5, which well represents the free-energy basin associated with the starting cryo-EM protein conformation.”

2) There is no interpretation about the strength of the couplings. Is it not relevant to compare their averages in order to determine optimal NPQ channels? I am sure it is.

Author's Reply: We agree with the Reviewer that the coupling strength is surely relevant to determine optimal NPQ channels within the complex. According to Lapillo et al. and Fox et al., the coupling strength observed here indicate that the present system is quenched. Furthermore, according to Lapillo

et al., the removal of the most coupled Car-Chl pair within each site (L1 and L2) is not sufficient to lower the NPQ level. This is relevant, because experimentally the complex is able to switch between quenched and unquenched states. We have added a section in the main article commenting on this:

“We note that, according to a previous study by some of us³⁹ where a kinetic model of EET was applied to CP29, these coupling values are compatible with a quenched conformation of the antenna. We can therefore conclude that, within this Coulomb approximation of the coupling, the two most coupled Car-Chl pairs in CP29 establish an efficient quenching channel that cannot be modulated by the protein environment.”

3) Authors must clearly explain their hypothesis as to what makes a more efficient EET. Is it a more variable (larger distribution of couplings, coulomb or short range) or a less sensitive distribution? What about the strength of the average couplings? I am failing to make the connection between sensitivity and variability and the efficiency in quenching. This is crucial, authors must lay out their hypothesis from the start. Besides, none of the analysis considers the resonances needed between the excited states of Chls and the S1 state of Car.

Author’s Reply: We thank the Reviewer for helping us explain better our reasoning. Up to now, models have appeared in the literature that predict an efficient quenching channel (refs 28,53,64,69,70 in the revised manuscript). However, an efficient quenching channel is not enough to explain the regulation of NPQ.

We have added a paragraph at the beginning of the Discussion section explaining the difference between an efficient quenching channel and one that has the further (desired) property of being tunable by the protein conformation. Only the latter can explain the onset/termination of NPQ that follows a conformational switch of the antenna. The paragraph is the following:

“It is generally accepted that conformational changes of LHCs are responsible for their switching between a light-harvesting and a quenched state. Furthermore, there have been several studies suggesting that Chl-Car EET based on a Coulomb approximation of the coupling can be used to explain the quenching^{28,53,64,69,70}. However, there seems to be a difficulty in understanding how the quenching efficiency can be regulated within the dynamic environment of these antennas.³⁹ Indeed, albeit an efficient quenching channel is mandatory for NPQ to work, this same channel should be switched on and off by different protein conformations, allowing plants to dissipate or collect light in response to external stimuli. What is then the connection between protein conformational changes and the regulation of NPQ?”

4) When I look at the couplings for Lut-a612 I do see that couplings are insensitive with respect to all possible cluster motions. I find this quite interesting. It says that no matter how the protein moves at room temperature, the Lut-a612 always delivers the same coupling. But from what I read, authors seem to imply that this is in detriment of assigning that pair as a quenching channel because it cannot be modulated: “Thus, the differences observed in Figure 3d could give rise to significant variations in the total coupling. This observation, combined with the insensitivity of the Coulomb interaction observed in L1 (Figure 3a), suggests that a putative Lut-a612 quenching channel can only be modulated by controlling the short-range coupling, rather than by altering the Coulomb coupling” I do not see the logic here.

Author's Reply: We have added a paragraph in the Results section explaining that, albeit this value of the Coulomb coupling indicates that the quenching channel is efficient, there is still the problem of how this channel can be modulated. In fact, it appears that a Coulomb approximation of the EET coupling indicates that the protein conformation cannot modulate this interaction. This paragraph is reported above (point n. 3). We have additionally clarified this in the Discussion section:

"This demonstrates that the Coulomb contribution to the EET coupling results in a quenching channel that cannot be tuned by the LHC conformation, indicating that this popular model is not sufficient to explain the NPQ regulation but two new aspects must be added. The first is the inclusion of short-range effects, which become important for the closely associated Car-Chl pairs such as Lut-a612 in L1 and Vio-a603 in L2. Indeed, our estimates of the short-range effects indicate that only adding this contribution to the coupling the Car-Chl interaction can become more sensitive to the protein conformation, thus linking the modulation of the quenching efficiency to the dynamic nature of LHCs. The second aspect to consider is the effect of the protein conformation on the Car internal geometry. In fact, our results indicate that both Lut and Vio undergo geometrical distortions in the respective sites, which are reflected in a significant tuning of the S1 energy and, consequently, of the overall dynamics of the antenna complex. This indicates that the control of the Car geometry is one way for the protein to tune the quenching efficiency⁴⁵, without significantly altering the Chl network."

5) Authors use the overlap as a correlated proxy to the short-range coupling, and mention that a 10 Å reduction is related to an order of magnitude drop in the coupling. This comes from a previous work by the authors where they analyzed triplet state couplings. It is not clear that this same correlation applies to singlet state couplings. Please elaborate on possible similarities/differences. Let's say that a similar correlation exists for the singlets. Are authors claiming that protein motions can access short-range couplings that are much larger than the coulomb couplings?

Authors' Reply: Following the Reviewer's suggestion, we have considered another way to compute the short-range coupling for singlet states. We have computed the short-range contribution to the EET coupling between the S₂ and Q_y states and analyzed its dependence on the overlap parameter. We have found an analogous dependence to the TET coupling and again an order of magnitude comparable to the S₁-Q_y Coulomb coupling. We have added a full section to the Supplementary Information explaining how we have conducted the QM/MMPol short-range analysis. We have rewritten the sections regarding the short-range interactions of the main article as follows:

The Coulomb approximation to the EET coupling is certainly valid the energy transfer is between bright states; however, this is not the case here where the dark S₁ state of Cars is involved. In these cases, and even more in triplet energy transfer (TET) where the Coulomb coupling is zero, short-range terms play a role.⁵⁷⁻⁵⁹ Unfortunately, short-range terms are difficult to compute, as they involve charge-transfer configurations⁶⁰. Moreover, here the difficulty of the calculations is further increased due to the lack of an established QM method for accurately describing these interactions in the case of the dark S₁ state.⁶¹

However, if we recognize that the short-range terms are strongly dependent on the overlap of electron densities of the pigments, we can try to get a rough estimate by computing the short-range contribution for the coupling between Chl Q_y and the bright state of Cars (S₂) which is instead accurately described by Time-Dependent DFT approaches (see Supplementary Methods). Moreover, in a previous study of some of us,⁶² we showed that the Car-Chl TET couplings in CP29

are strongly sensitive to the overlap parameter, a geometrical approximation of the electronic overlap in terms of rigid spheres centered on the atoms of the Car-Chl pair (see Supplementary Methods). Indeed, the short-range couplings calculated for the S_2/Q_y energy transfer are similar in magnitude to the ones previously calculated for TET (and to the Coulomb S_1/Q_y couplings reported in Figure 4)), and they show a similar dependence on the overlap parameter as shown in Supplementary Figure 4.

6) At the top of page 7, do authors mean the Coulomb coupling of Vio-a602 instead of Vio-a603? Right after, authors talk about the stromal ring of Vio, but the stromal ring of Vio appears to interact mainly with a602.

Authors' Reply: Indeed, the movements of the N-terminal do have an effect on the interaction of both Vio-a602 and Vio-a603. The former is more affected than the latter as it lies close to the N-terminal. We have commented on the conformational freedom of the stromal side of Vio, and clarified its influence on Chl a602, by adding the following sentence to the Results section:

“On the other hand, the Vio-Chla602 interaction is affected by the conformational freedom of Vio in the stromal side (Figure 4c).”

7) Speaking about variability of couplings, would it not have been equally important to analyze the variability of their DFT/MRCI S_1 states? After all, if there is no resonance between the S_1 state of cars and the Q_y of Chl's there is no EET.

Authors' Reply: We agree with the Reviewer that the variability of S_1 energy is an important factor for the regulation of quenching. Unfortunately, performing DFT-MRCI calculations for many structures along the PT-WTE dynamics is not computationally feasible. Following the suggestion of the Reviewer, we have performed Semiempirical CI (SECI) calculations to obtain the S_1 energy of the carotenoids using the QM/MM optimized geometries previously obtained to check the reliability of the MM force field (see answer 1 to Reviewer 2). These calculations show that the S_1 energy differs among the clusters and is dependent on the carotenoid conformation. We have added a new Figure (Figure 3 of the revised manuscript) and a full section to the Supplementary Information, explaining how we have computed the S_1 energy for both carotenoids. We have added a paragraph to the main article commenting on these results:

“We have further investigated the dependence on the geometry of the Car S_1 state by means of semiempirical CI (SECI) calculations, which have been recently shown to reasonably describe the electronic structure of keto-carotenoids⁵². Indeed, SECI calculations confirm the tunability of the Car S_1 energy, which is different in different clusters (Figure 3c and Supplementary Figure 5a,b) and further depends on the s-cis/s-trans conformation (Figure 3b and Supplementary Figure 5c,d), thereby confirming the impact of the protein dynamics on the electronic structure of the embedded carotenoids.”

8) Figure 7b shows the free energy map as function of collective variables that produce a local minimum at an open conformation. However, it looks that the open conformation is 10 kcal/mol higher than the CryoEM, how could this conformation be present at equilibrium?

Author's Reply: Indeed, the free energy surface suggests that the "Open" conformation is not thermodynamically favored in isolated CP29. However, the environment of the thylakoid membrane and the protein-protein interactions within PSII may change the relative stability of the conformations, possibly bringing the Open conformation closer to the Cryo-EM one. We have commented on this within the Results section and in the Conclusions:

"We estimate that this conformation is separated by several kcal/mol from the cryo-EM one (Figure 5b, Supplementary Figure 12), and additional stabilizing factors would then be required to make it accessible at room temperature."

"It can thus be likely possible that those triggers change the relative stability of the different conformers (including the metastable 'open' one), for instance by promoting protein-protein interactions."

On the other hand, the couplings obtained on the Open conformation confirm our conclusions made on PT-WTE results: the Coulomb couplings in the L1 site are minimally sensitive to even large changes in conformation, and in turn they cannot explain how the quenching can be tuned by the protein.

REVIEWERS' COMMENTS

Reviewer #1 (Remarks to the Author):

The authors have addressed adequately all issues raised by the reviewers, therefore I recommend publication in Nat. Commun.

Reviewer #2 (Remarks to the Author):

The authors have sufficiently addressed all points raised on the original submission. They include new results from additional calculations, and present an improved manuscript both in content and in presentation thanks to important clarifications. I have no further comments on the revised version.

Reviewer #3 (Remarks to the Author):

In the revised manuscript, authors have more clearly stated their hypothesis and have defined what makes an efficient quenching channel.

Authors have further computed the short-range contribution to the EET coupling between the S2 and Qy states and analyzed its dependence on the overlap parameter, in agreement with their previous triplet state coupling results. Other clarifications and format changes have improved the manuscript. I recommend publication as is.

J. A. Gascón